# Effect of Supplementary Lighting Duration on Growth and Activity of Antioxidant Enzymes in Grafted Watermelon Seedlings

**Hao Wei [1],[†]** , **Mengzhao Wang [1],[†] and Byoung Ryong Jeong [1],[2],[3],\***

[1] Department of Horticulture, Division of Applied Life Science (BK21 Plus Program), Graduate School of Gyeongsang National University, Jinju 52828, Korea; oahiew@gmail.com (H.W.); mengzhao.waung@gmail.com (M.W.)

[2] Institute of Agriculture & Life Science, Gyeongsang National University, Jinju 52828, Korea

[3] Research Institute of Life Science, Gyeongsang National University, Jinju 52828, Korea

\* Correspondence: brjeong@gnu.ac.kr; Tel.: +82-010-6751-5489

[†] These authors contributed equally to this work.

**Abstract:** Insufficient exposure to light in the winter may result in a longer production periods and lower quality of seedlings in greenhouses for plug growers. Supplementary artificial lighting to plug seedlings may be one solution to this problem. The objective of this study was to assess the effects of the duration of the supplementary light on the growth and development of two watermelon cultivars, 'Speed' and 'Sambok Honey' grafted onto 'RS-Dongjanggun' bottle gourd rootstocks (*Lagenaria siceraria* Stanld). Seedlings were grown for 10 days in a glasshouse with an average daily natural light intensity of 340 $\mu mol \cdot m^{-2} \cdot s^{-1}$ photosynthetic photon flux density (PPFD) and daily supplementary lighting of 8, 12 or 16 h from mixed LEDs ($W_1R_2B_1$, chip ratio of white:red:blue = 1:2:1) at a light intensity of 100 $\mu mol \cdot m^{-2} \cdot s^{-1}$ PPFD, a group without supplementary light was set as the control (CK). The culture environment in a glasshouse had 25/15 °C day/night temperatures, an 85 ± 5% relative humidity, and a natural photoperiod of 8 h. The results showed that all the growth and development parameters of seedlings grown with supplementary light were significantly greater than those without supplementary light (CK). The 12 and 16 h supplementary light resulted in greater growth and development parameters than the 8 h supplementary light did. The same trend was also found with the indexes that reflect the quality of the seedlings, such as the dry weight ratio of the shoot and root, total biomass, dry weight to height ratio of scions, and specific leaf weight. The 12 h and 16 h light supplements resulted in greater Dickson's quality indexes compared to the 8 h supplementary light, and the 12 h supplementary light showed the greatest use efficiency of the supplementary light. 16 h of daily supplementary light significantly increased the $H_2O_2$ content and the antioxidant enzyme activities in seedlings compared to the other treatments. This indicated that 16 h of supplementary light led to certain stresses in watermelon seedlings. In conclusion, considering the energy consumption, 12 h of supplementary light was the most efficient in improving the quality of the two cultivars of grafted watermelon plug seedlings.

**Keywords:** biomass; DLI; LEDs; light use efficiency

## 1. Introduction

Watermelon (*Citrullus vulgaris*) is a highly popular fruit vegetable with a great economic value, widely grown all over the world. In South Korea, 99% of watermelon seedlings are grafted. Most of them are produced by professional seedling companies. With the development of the plug seedling

industry, high quality plug seedlings are in great demand. Thus, an efficient production of high-quality plug seedlings has become a focus of ongoing research.

Light is the primary environmental factor for the growth, development, pigmentation, and morphogenesis of plants [1–4]. The low natural light level during the rainy season and winter time is a limiting factor for vegetable production in greenhouses in northern regions [5]. Many research has shown the effects of low light intensities on the growth and development of plants [6–8]. Under a weak light condition, chlorophyll, carotenoid and soluble sugar contents in the leaves of towel gourd decrease significantly. The vegetative growth parameters of plants such as the height, root development and chlorophyll (SPAD) levels in rice leaves were affected by weak light [9]. The chlorophyll contents were also found to be affected by the light intensity in some cultivars of *Lilium* Oriental hybrids [10].

Plants convert inorganic substances into organic compounds through photosynthesis. Efficiently producing high-quality products has always been a common goal of growers, especially in crop production. The lack of light caused by the geographical location, climate, or outdated facilities will seriously prolong the production time of most crops, which directly results in low yield, high cost and poor quality [11–13]. Lately, growers have been paying more attention to 'light fertilization' than to water and nutrition fertilization. Artificial supplementary light is the best solution to the lack of light caused by the geographical location or climate [14–16]. Light is an environment-friendly energy, provided through electricity. Although the energy consumption required to provide light is high, light supplement is very valuable because of the profit of high-quality products with prices rising on many crops. Thus, efficient supplementation of light for such crops as propagated young plants reasonably has become a meaningful research topic.

In recent years, an increasing number of supplementary light technology using light emitting diodes (LEDs) has been used in agricultural production [17–20]. The improvement in the growth, yield, and quality of crops has been confirmed by many studies [6,21–23]. In addition, different light qualities also had different effects on the growth of cucumber and pepper [24–28]. Blue and red LEDs effectively promoted the development and accumulation of dry biomass in lettuce [29]. However, most studies investigated the effects of the quality and intensity of natural and supplementary lighting on the growth and yield of greenhouse vegetables during the winter. Only a few studies investigate the effects of the duration of the supplementary lighting on vegetables, especially grafted plug seedlings during, and right after, the graft healing period, which limited the practical application of supplementary lighting to a certain extent.

In South Korea, the lack of sunlight during the winter is one of the constraints affecting the production of vegetable and fruit crops. Especially when the natural light passes through the covering material of greenhouse, the light is further reduced. We had recorded the natural light condition in a venlo type glasshouse with a light transmittance of about 65% from December 15th–25th of 2018. The average daily light intensity was only 340 $\mu$mol·m$^{-2}$·s$^{-1}$ photosynthetic photon flux density (PPFD) (Daily light integral (DLI) = 9.79 mol·m$^{-2}$·d$^{-1}$) (unpublished data). This was far from being sufficient for the growth of the grafted watermelon seedlings. The low light environment will delay the delivery and reduce the quality of the seedlings compared to other sunny seasons. Artificial light sources can be used to facilitate graft healing during grafting. After grafting healing, during the domestication period the assimilation of seedlings can also be increased by supplementary lighting so as to improve the quality of seedlings and shorten the production time.

In previous studies, the optimal quality and intensity of supplementary lighting were determined [30–34]. Based on these previous studies, three durations of supplementary light (8, 12, 16 h/day) at 100 $\mu$mol·m$^{-2}$·s$^{-1}$ PPFD using mixed LEDs ($W_1R_2B_1$) were tested to investigate the effects of different durations of supplementary lighting on the quality of grafted watermelon plug seedlings grown in a glasshouse after the graft healing period.

## 2. Materials and Methods

### 2.1. Plant Materials

The commercially available grafted seedlings of two cultivars of watermelon (*Citrullus vulgaris* Schrad.), 'Speed' (Nongwoo Bio Co., Ltd., Suwon, Republic of Korea) and 'Sambok Honey' (Farm Hannong Co., Ltd., Seoul, Republic of Korea), were grafted onto 'RS Dongjanggun' (Syngenta Korea Co., Ltd., Seoul, Republic of Korea) bottle gourd (*Lagenaria siceraria* Standl.) rootstocks by using splice grafting. The seedlings were all grown in 40 square-cell plug trays containing a commercial medium (Super Mix, NongKyung Co., Jincheon, Republic of Korea) in a venlo type glasshouse. The culture environment in the glasshouse had 25/15 °C day/night temperatures, an 85 ± 5% relative humidity, and a natural photoperiod of 8 h. These two scion genotypes grafted on the rootstock were selected as these two cultivars are extensively used for grafting purposes in the Republic of Korea because of their high compatibility with this rootstock, high ratio of seed germination, and great fruit quality.

### 2.2. Light Treatments and Culture Environment

Light with an intensity of 100 $\mu$mol·m$^{-2}$·s$^{-1}$ PPFD from LEDs (W$_1$R$_2$B$_1$, chip ratio of white:red:blue = 1:2:1) (Custom made, SungKwang LED Co., Ltd., Incheon, Republic of Korea) was provided as the only supplementary light source (Figure 1). One LEDs bar (150 × 3 cm) was series connected by 200 LED chips and the power of each LED chip was 2 watts. Well-healed grafted seedlings were taken out from the healing room and then put immediately into a venlo type glasshouse (35°09' N, 128°05' E, Jinju, South Korea) with a daily average light intensity of 340 $\mu$mol·m$^{-2}$·s$^{-1}$ PPFD coming from the sun and 100 $\mu$mol·m$^{-2}$·s$^{-1}$ PPFD of supplementary lighting from the mixed LEDs (W$_1$R$_2$B$_1$). After recording the natural light intensity during the entire experimental period (December 15th–25th, 2018), the daily average light intensity was obtained. The supplementary light intensity was set at night time with a fixed value of 100 ± 5 $\mu$mol·m$^{-2}$·s$^{-1}$ PPFD by a dimmer after measuring at multi-point averages in each treatment area. Light intensity was measured by a probe of quantum radiation (FLA 623 PS, ALMEMO, Holzkirchen, Germany) at the level of the top leaf of the seedlings. The daily supplementary light durations were set as either 8 (08:00–16:00, Supplementary DLI = 2.88 mol·m$^{-2}$·d$^{-1}$ ), 12 (06:00–18:00, Supplementary DLI = 4.32 mol·m$^{-2}$·d$^{-1}$), or 16 (04:00–20:00, Supplementary DLI = 5.76 mol·m$^{-2}$·d$^{-1}$) hours. The 12:00 of the day was set as a median time point so that the supplementary lighting time in the morning and that in the afternoon were the same. In addition, a group without supplementary light was set as the control (CK, Supplementary DLI = 0.00 mol·m$^{-2}$·d$^{-1}$). All LEDs were fixed to 1.0 m above the seedling bench (The distance from the top leaf of the seedlings was about 85 cm), and each bench was treated with one supplementary light duration. There was enough space between the benches to ensure that the treatments did not interfere with each other. The control (CK) was set with the same LEDs with supplementary lighting treatments, but not turned on to balance the shadow of natural light. The automatic control system of the glasshouse recorded the temperature and humidity during the experiment period. The culture environment had 25/15 °C day/night temperatures and an 80 ± 5% relative humidity, with a natural photoperiod of 8 h. The definition of day and night were based on the sunrise and sun set during the experimental period.

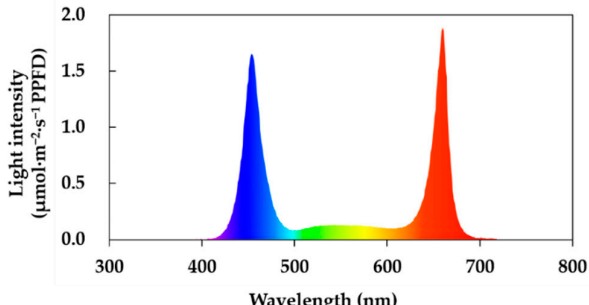

**Figure 1.** The spectrum of the supplementary light source (mixed LEDs: $W_1R_2B_1$) used in the experiment: W, cold white LED (peak wavelength at 452 and 561 nm); R, red LED (peak wavelength at 659 nm); and B, blue LED (peak wavelength at 452 nm).The spectral distributions were measured with a portable spectroradiometer (Spectra Light ILT 950, International Light Technologies, Inc., Peabody, MA, USA).

*2.3. Measurement of Soluble Sugars, Starch, and Soluble Proteins*

The contents of soluble sugars and starch extracts were measured by the anthrone colorimetric method as described by Xue et al. [35]. After homogenized the 0.2 g of fresh leaf samples were extracted in 20 mL distilled water for 30 min at 100 °C. After centrifugation at 6500 rpm for 15 min, the solution was filtered into a 100 mL volumetric flask for the assay of soluble sugars, and the solution was brought up to 100 mL with distilled water. The insoluble substance was re-extracted in 20 mL distilled water for 15 min at 100 °C, then mixed with perchloric acid (2 mL, 52%) for another 15 min at 100 °C. Afterwards, the solution was filtered into a 100 mL volumetric flask and the volume of the solution was brought up to 100 mL with distilled water for the measurement of starch.

For the measurements of soluble sugars extract, 0.2 mL of soluble sugars was mixed with distilled water (1.8 mL), anthrone (0.5 mL, 2%), and sulfuric acid (5.0 mL, 98%). After a 10 min incubation at 100 °C, the absorbance at 630 nm was recorded with a UV-spectrophotometer (Libra S22, Biochrom Ltd., Cambridge, UK).

For the measurement of starch extract, 1.0 mL of soluble starch, 1.0 mL distilled water, 1.0 mL anthrone, and 5 mL of 98% sulfuric acid have been mixed. After the solution was incubated at 100 °C for 10 min, the absorbance at 430 nm was recorded with a UV-spectrophotometer as described before. The calibration curves of soluble sugars and starch were made with the respective standard solutions.

The total soluble proteins were determined by Bradford's method [36]. The 0.1 g of fresh leaf samples were ground using liquid nitrogen. After fully ground, the leaf powder has been extracted with a 1.5 mL sodium phosphate buffer for 10 min. The homogenate was centrifuged at 13,000 rpm for 20 min at 4 °C and the supernatant was transferred to a new tube for assays. For protein estimation, 50 μL of the supernatant was mixed with 1450 μL of the Braford reagent and incubated for 10 min at room temperature. The absorbance at 595 nm was recorded with a UV-spectrophotometer as described before. The calibration curve of proteins was made using bovine serum albumin (BSA) as the standard.

*2.4. Measurement of Hydrogen Peroxide*

The spectrophotometric determination of $H_2O_2$ was carried out according to the method described by Christou et al. [37]. Leaf samples (0.1 g) were homogenized in 1 mL of 0.1% trichloroaceticacid (TCA) and centrifuged at 10,000 rpm for 15 min. Then, 0.5 mL of the supernatant was mixed with a 10 mM phosphate buffer (0.5 mL, pH 7.0) and 1 mL potassium iodide (1M). The mixture was incubated at room temperature for 30 min, after which the absorbance was measured at 390 nm. The $H_2O_2$ content was determined from the standard calibration curve.

### 2.5. Measurement of Activities of Antioxidant Enzymes

For the analysis of antioxidant enzymes, 0.1 g fresh leaf samples were homogenized in a 1.5 mL ice-cold 50 mM phosphate buffer (pH 7.0) containing 1 mM EDTA, 0.05% triton X-100, and 1 mM polyvinylpyrrolidone (PVP). The extracts were centrifuged at 13,000 rpm for 20 min at 4 °C and the supernatant was employed for the analysis of enzyme activities. The superoxide dismutase (SOD) activity was estimated by following the nitro blue tetrazolium (NBT) inhibition methods according to the protocol of Giannopolitis and Ries [38]. The activity of the catalase (CAT) enzyme was measured based on the method of Cakmak and Marschner [39]. The guaiacol peroxidase (GPX) activity was determined based on the amount of enzyme required for the formation of tetraguaiacol per minute, following the methods of Shah et al. [40]. The ascorbate peroxidase (APX) activity was assayed following the methods of Nakano and Asada [41].

### 2.6. Measurements of Growth Parameters and Data Collection

After 10 days of treatment, the seedlings treated with supplementary light all reached the standard of transplanting. The transplanting standard of watermelon seedling in South Korean is when the seedling has 2-3 true leaves, and a well-developed root system so that the root-ball can be pulled out from the plug tray. Thus, the growth parameters such as length, fresh and dry weight, and diameter of shoot, number, area, and chlorophyll content (SPAD) of leaf, and length, fresh and dry weight of root were measured. From the graft union to the top and base of the seedling have been measured as scion and rootstock length, respectively. Both fresh and dry (dry in oven with 70 °C for 72 h) weights were measured by electronic balance (ENTRIS224I-1S, Sartorius, Goettingen, Germany). The scion and rootstock diameter were measured by a digital vernier caliper (CD-20CPX, Mitutoyo, Kawasaki, Japan) at 1 cm above and below the graft union. The fully unfolded leaf was considered 1 leaf. Leaf area was measured by leaf area measuring instrument (LI-3000, LI-COR Inc., Lincoln, NE, USA). Leaf chlorophyll contents were measured with fully expanded mature leaf by SPAD instrument (SPAD-502, Konica Minolta Inc., Tokyo, Japan). Each leaf was averaged after three measurements. Root length was the length from the root base to the tip of longest root.

After the parameters were measured, the dry weight ratio of the shoot and root, total biomass, scion dry weight to height ratio, specific leaf weight, Dickson's quality index (DQI), and supplementary light use efficiency were also calculated. The DQI was calculated as shown below [42–45]:

$$\text{Dickson's quality index (DQI)} = \frac{\text{Total dry weight of seedling (g)}}{\dfrac{\text{Height (mm)}}{\text{Stem diameter (mm)}} + \dfrac{\text{Shoot dry weight(g)}}{\text{Root dry weight (g)}}} \tag{1}$$

In order to show the use efficiency of the supplementary light by seedlings with the different durations of supplementary lighting, the increase in the seedling biomass after 1 mol of the photon was excited from the source of the supplementary light has been calculated as follows:

$$\begin{aligned}&\text{S} - \text{light use efficiency}\\ &= \frac{\Delta\text{Biomass (Treatment biomass–CK biomass) (g)}}{\text{S–light duration (s)} \times \text{Leaf area (m}^2) \times \text{S–light intensity } (\mu\text{mol·m}^{-2}\text{·s}^{-1})}\end{aligned} \tag{2}$$

(*S-, Supplementary; Treatment biomass, the biomass after 10 days of supplementary light treatment; CK biomass, the biomass of the control after 10 days.)

A randomized complete block designed with 3 replications and 9 seedlings in each replication was employed in this experiment. Each treatment had 3 plug trays, each tray as a replication (1 block) and each plant as an observational unit. The treatment locations in a controlled environment were randomly laid out to minimize location effects. The collected data were analyzed for statistical significance with the SAS (Statistical Analysis System, V. 9.1, Cary, NC, USA) program. The experimental results were subjected to an analysis of variance (ANOVA) and Tukey's multiple range test.

## 3. Results

### 3.1. Growth and Developmental Parameters

As shown in Table 1, after ten days of treatment, all seedlings grown with supplementary lighting had longer scions than those in CK. For 'Speed', there were no differences observed in the scion length between seedlings grown with 12 h of supplementary light and 16 h of supplementary light. However, shorter scions were found for seedlings grown with 8 h of supplementary light. For 'Sambok Honey', there were no significant differences in the scion length between seedlings grown with 8, 12 and 16 h of supplementary light. No significant differences were observed in the rootstock length among all seedlings. The two cultivars of watermelon seedlings exposed to 12 and 16 h of supplementary light had a similar fresh weight of scions, which was higher than that of seedlings exposed to 8 h of supplementary light, while seedlings in CK had the lowest scion fresh weight. However, within each cultivar, no significant differences were observed in the fresh weight of rootstocks between seedlings grown with 8, 12 and 16 h of supplementary light. The 12 and 16 h of supplementary light resulted in similar and greatest dry weights of the scions and rootstocks, while seedlings in CK had the lowest values. The diameter of the scions and rootstocks of 'Speed' and 'Sambok Honey' increased with the duration of the supplementary lighting, but there were no significant differences in the diameter of the scions and rootstocks between seedlings grown with 12 and 16 h of supplementary light. For 'Speed', seedlings grown with supplementary light had no significant differences in the number of leaves, but 12 and 16 h gave largest number in 'Sambok Honey', The 12 and 16 h gave the greatest leaf area for 'Speed', but not 'Sambok Honey'. For the chlorophyll content, the 12 h treatment gave no significant differences with 16 h, but higher than 8 h in 'Speed'. There were no significant differences among all supplementary light treatments in 'Sambok Honey'. However, all supplementary light treatments showed higher chlorophyll content than CK in both cultivars. There were no significant differences in the root length among 'Speed' seedlings exposed to the different durations of supplementary light. 'Sambok Honey' seedlings had the shortest roots when grown without any supplementary light. Both 'Speed' and 'Sambok Honey' seedlings had similar trends of fresh and dry weights of the roots in response to the supplementary light duration. Seedlings grown with 12 and 16 h of supplementary light had the greatest root fresh and dry weights without a significant difference between them, followed by those grown with 8 h of supplementary light and then the control.

In general, most growth and developmental parameters of seedlings treated with supplementary lighting were greater than those in the control. There was almost no difference between the growth parameters of seedlings grown with 12 h and 16 h of supplementary light, which were better compared to the parameters of seedlings grown with 8 h of supplementary light. The two cultivars 'Speed' and 'Sambok Honey' had no interaction effects with the different treatments.

**Table 1.** The effects of the supplementary light duration on the growth and development of two cultivars of watermelon 'Speed' and 'Sambok Honey' grafted onto 'RS  Dongjanggun' bottle gourd rootstocks after 10 days of treatment.

| Cultivar [A] | Supplementary Light Duration (h) [B] | Shoot | | | | | | | | Leaf | | | Root | | |
|---|---|---|---|---|---|---|---|---|---|---|---|---|---|---|---|
| | | Length (cm) | | Fresh Weight (g) | | Dry Weight (g) | | Diameter (mm) | | No. | Area (cm²) | Chlorophyll Content (SPAD) | Length (cm) | Fresh Weight (g) | Dry Weight (g) |
| | | Scion | Rootstock | Scion | Rootstock | Scion | Rootstock | Scion | Rootstock | | | | | | |
| 'Speed' | CK | 8.0 c$^z$ | 6.1 | 0.91 d | 2.35 cd | 0.05 d | 0.10 c | 3.10 c | 5.15 cd | 3.5 cd | 9.1 c | 36.63 c | 14.6 bc | 0.42 d | 0.03 c |
| | 8 | 12.1 b | 6.3 | 2.22 c | 2.59 abc | 0.14 c | 0.13 b | 3.85 ab | 5.26 bcd | 4.0 bc | 19.3 b | 51.53 b | 14.6 bc | 1.46 b | 0.04 bc |
| | 12 | 13.7 a | 5.9 | 3.75 a | 2.82 ab | 0.26 a | 0.16 a | 4.16 a | 5.68 a | 4.5 ab | 24.0 a | 58.87 a | 17.1 ab | 2.20 a | 0.11 a |
| | 16 | 14.0 a | 6.3 | 3.75 a | 2.85 a | 0.27 a | 0.16 a | 4.16 a | 5.66 a | 4.5 ab | 24.2 a | 53.22 ab | 14.1 bc | 2.18 a | 0.12 a |
| 'Sambok Honey' | CK | 8.3 c | 6.1 | 0.90 d | 2.25 d | 0.05 d | 0.11 c | 3.49 bc | 5.08 d | 3.2 d | 11.5 c | 40.40 c | 13.0 c | 0.59 d | 0.02 c |
| | 8 | 10.7 b | 6.2 | 2.16 c | 2.52 bcd | 0.18 bc | 0.13 b | 3.71 ab | 5.56 ab | 3.8 bcd | 22.8 ab | 50.85 b | 15.6 abc | 1.06 c | 0.05 b |
| | 12 | 11.8 b | 6.4 | 3.23 ab | 2.52 bcd | 0.22 ab | 0.16 a | 3.88 ab | 5.50 abc | 4.8 a | 22.1 ab | 56.33 ab | 18.1 a | 2.24 a | 0.12 a |
| | 16 | 12.0 b | 6.1 | 3.18 b | 2.53 bcd | 0.23 a | 0.17 a | 4.09 a | 5.71 a | 5.0 a | 22.2 ab | 56.47ab | 18.3 a | 2.23 a | 0.13 a |
| F-test | A | **$^y$ | NS | * | ** | NS | NS | NS | NS | NS | NS | NS | NS | NS | NS |
| | B | *** | NS | *** | *** | *** | *** | *** | *** | *** | .*** | *** | ** | *** | *** |
| | A*B | NS | NS | NS | NS | NS | NS | NS | NS | NS | NS | NS | NS | NS | NS |

$^z$Mean separation within columns by Tukey's multiple range test at $p = 0.05$. $^y$NS, *, **, ***, No significant or significant at $p = 0.05$, 0.01, or 0.001, respectively.

### 3.2. Seedling Quality Index

Existing growth parameters were calculated and analyzed to compare the quality differences of the grafted watermelon seedlings (Figure 2). For 'Speed', seedlings grown in CK had the highest dry weight ratio of the shoot and root, and seedlings grown with 8, 12 and 16 h of supplementary light showed no significant differences in the dry weight ratio of the shoot and root. For 'Sambok Honey' seedlings, CK and 8 h of supplementary light resulted in a higher dry weight ratio of the shoot and root than 12 and 16 h of supplementary light did. The two cultivars exhibited a similar trend with the biomass, where the biomass increased with the increase in the duration of the supplementary light, but without a significant difference between 12 h and 16 h of supplementary light. For 'Speed', seedlings grown with 12 and 16 h of supplementary light had the greatest dry weight to height ratio of the scion, followed by those grown with 8 h of supplementary light and in CK. For 'Sambok Honey', CK similarly resulted in the lowest dry weight to height ratio of the scion, but no significant differences were observed in the dry weight to height ratio of the scion in among seedlings grown with 8, 12 and 16 h of supplementary light. 'Sambok Honey' seedlings grown with 12, 16, 8, and 0 h of supplementary light had the greatest to lowest specific leaf weight. 'Speed' seedlings exhibited no significant differences in the specific leaf weight among those grown with 8, 12 and 16 h of supplementary light, while seedlings in CK had the lowest specific leaf weight.

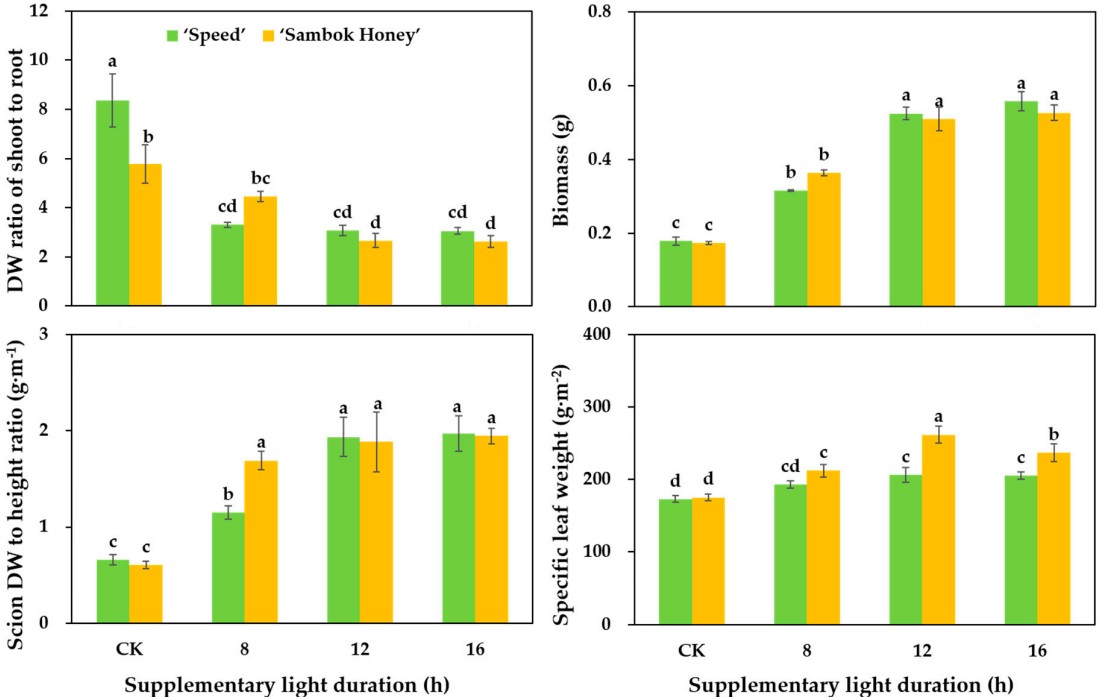

**Figure 2.** The effects of the supplementary light duration on the dry weight ratio of the shoot and root, biomass, dry weight to height ratio of the scion, and specific leaf weight of two cultivars of watermelon, 'Speed' and 'Sambok Honey', grafted onto 'RS Dongjanggun' bottle gourd rootstocks, after 10 days of supplementary light treatments. The error bars represent the SEs of the biological replicates ($n = 3$). Mean separation within columns are significantly different according to Tukey's multiple range test at $p = 0.05$.

For Dickson's quality index, there was no significant difference between the two cultivars. Seedlings grown with 12 and 16 h of supplementary light had the greatest index, followed by those with 8 h and those in the CK. By calculating the use efficiency of the supplementary light, it was found that 12 h of supplementary light resulted in the highest light utilization rate in the two cultivars. 16 h of supplementary light had a lower use efficiency of the light compared to the 12 h. 'Speed' seedlings grown with 8 h of supplementary light had a lower use efficiency than those with 12 h, while no

significant differences were found in the use efficiency between 'Sambok Honey' seedlings grown with 8 and 12 h of supplementary light (Figure 3).

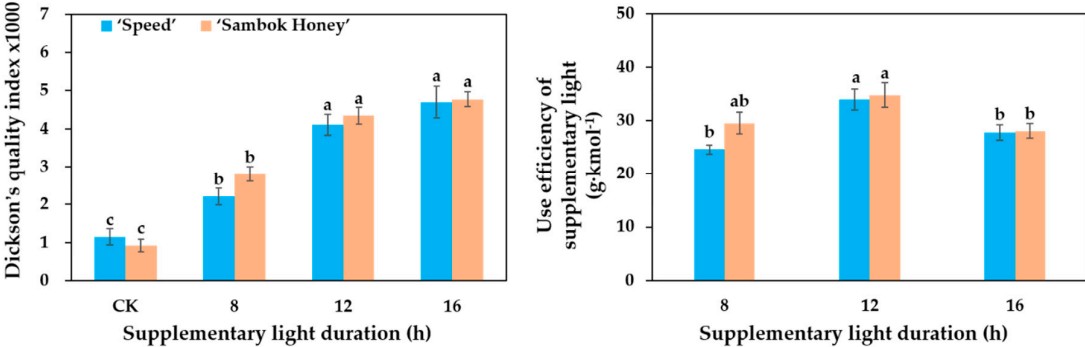

**Figure 3.** The effects of the supplementary light duration on the Dickson's quality index and use efficiency of the supplementary light of two cultivars of watermelon, 'Speed' and 'Sambok Honey', grafted onto 'RS Dongjanggun' bottle gourd rootstocks, after 10 days of supplementary light treatments. The error bars represent the SEs of the biological replicates ($n$ = 3). Mean separation within columns are significantly different according to Tukey's multiple range test at $p$ = 0.05.

### 3.3. Seedling Morphology

The morphology of the 'Speed' and 'Sambok Honey' seedlings affected by the different durations of the 10 day supplementary light treatments are shown in Figure 4. It is clear that the seedlings grown with 12 and 16 h of supplementary light had a higher biomass, were more compact, and looked healthier. The quality of the watermelon seedlings, especially the root ball formation, increased with the increase in the supplementary lighting duration; but there was little difference in the quality of seedlings grown with 12 and 16 h of supplementary lighting.

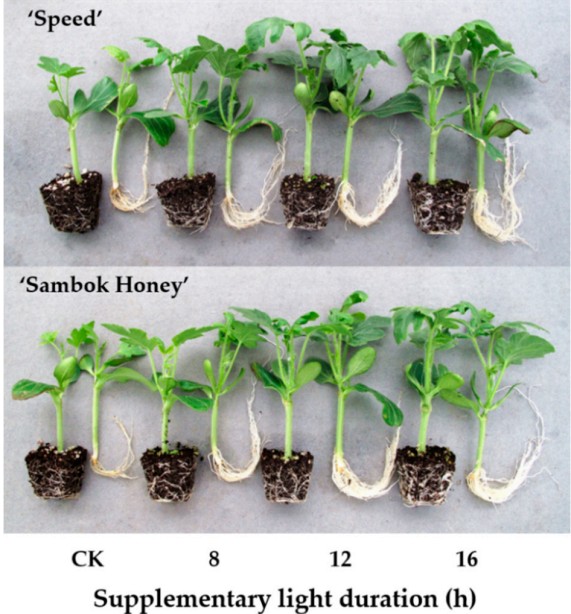

**Figure 4.** The effects of the supplementary light duration on the morphology of two cultivars of watermelon seedlings, 'Speed' and 'Sambok Honey', grafted onto 'RS Dongjanggun' bottle gourd rootstocks, after 10 days of supplementary light treatments.

### 3.4. Contents of Carbohydrates and Soluble Proteins

The contents of carbohydrates and soluble proteins in seedlings significantly increased with supplementary lighting through increased photosynthesis (Figure 5). 'Speed' seedlings grown with 12 and 16 h of supplementary lighting displayed no significant differences in the soluble sugars and starch contents, while seedlings grown with 8 h of supplementary light had lower levels of soluble sugars and starch, and seedlings in CK had the lowest levels of soluble sugars and starch. 'Sambok Honey' seedlings supplied with 12 and 16 h of supplementary light had the greatest starch content, followed by those with 8 h of supplementary light and in CK. Seedlings grown with supplementary light displayed no significant differences in the levels of soluble sugars, which were all higher than those of seedlings grown in CK. For the contents of soluble proteins, no significant differences were found among seedlings grown with supplementary light, but all of them had higher protein levels compared to those grown in CK.

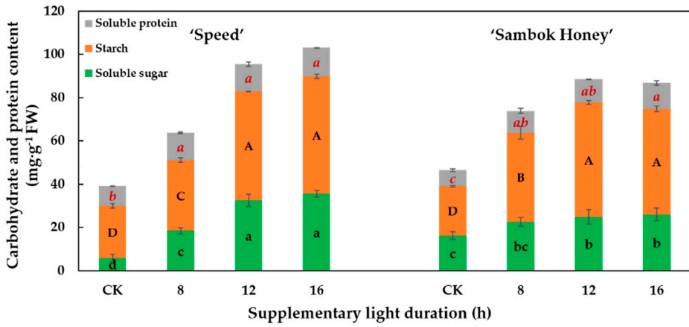

**Figure 5.** The effects of the supplementary light duration on the contents of soluble sugars, starch and soluble proteins in two cultivars of watermelon, 'Speed' and 'Sambok Honey', grafted onto 'RS Dongjanggun' bottle gourd rootstocks, after 10 days of supplementary light treatments. The error bars represent the SEs of the biological replicates (*n* = 3). Mean separation within columns are significantly different according to Tukey's multiple range test at *p* = 0.05.

### 3.5. Hydrogen Peroxide Content and Activities of Antioxidant Enzymes

The hydrogen peroxide ($H_2O_2$) content was measured and analyzed after 10 day treatments of supplementary light in the two cultivars of the grafted watermelon seedlings (Figure 6). For both cultivars, 16 h of supplementary lighting resulted in the highest $H_2O_2$ levels. 8 and 12 h of supplementary light led to the lowest $H_2O_2$ levels in 'Speed' seedlings, while 8 h of supplementary light resulted in the lowest $H_2O_2$ levels in 'Sambok Honey' seedlings, followed by 12 h and CK.

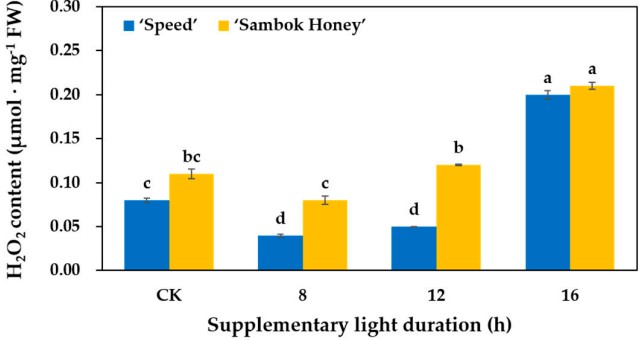

**Figure 6.** The effects of the supplementary light duration on the hydrogen peroxide content ($H_2O_2$) of two cultivars of watermelon seedlings, 'Speed' and 'Sambok Honey', grafted onto 'RS Dongjanggun' bottle gourd rootstocks, after 10 days of supplementary light treatments. The error bars represent the SEs of biological replicates (*n* = 3). Mean separation within columns are significantly different according to Tukey's multiple range test at *p* = 0.05.

The activities of four antioxidant enzymes were also measured and analyzed (Figure 7). The superoxide dismutase (SOD) activity increased with the increase in the supplementary light duration for 'Speed'. There was no significant difference between CK, 8, and 12 h treatments, but 16 h treatment showed significantly higher activity of SOD than other treatments for 'Sambok Honey'. The activity of catalase (CAT) was the highest in seedlings supplemented with 16 h of light, eacpet the 'Speed' in CK, other treatments were significantly lower than 16 h of treatment. Generally, 'Speed' seedlings had higher ascorbate peroxidase (APX) activities compared to 'Sambok Honey' seedlings, and 16 h of supplementary lighting resulted in the highest APX activities in both cultivars. Both cultivars had the highest guaiacol peroxidase (GPX) activities in seedlings grown with 16 h of supplementary light. 'Speed' seedlings had the lowest GPX activities with 12 h or no supplementary lighting, while 'Sambok Honey' seedlings had the lowest GPX activities with 8 h of supplementary light, and seedlings with 12 h or no supplementary light displayed similar intermediate GPX activity levels.

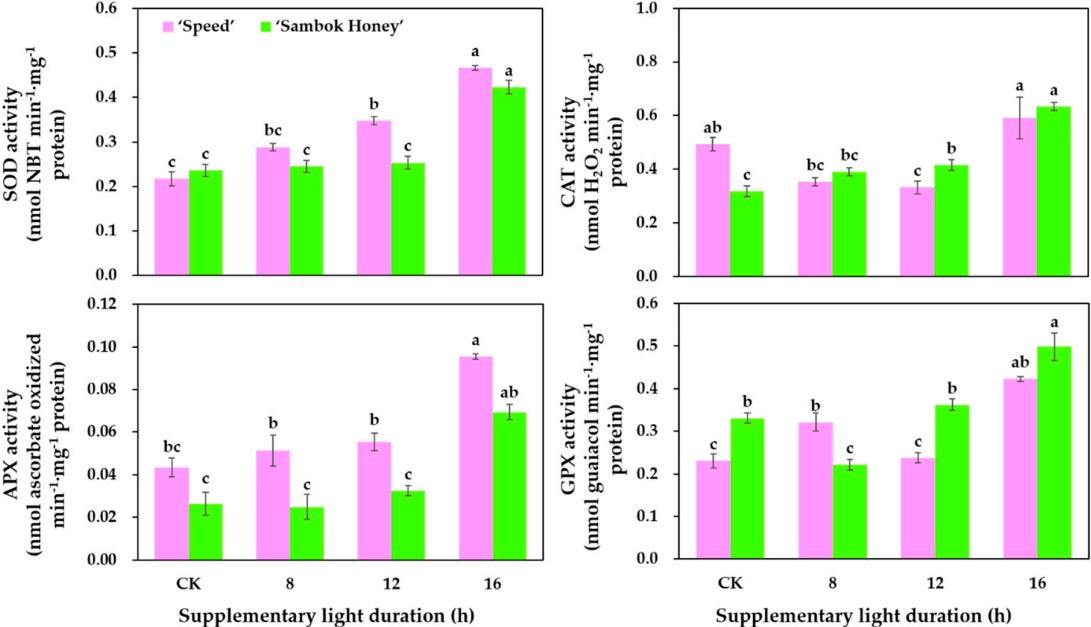

**Figure 7.** The effects of the supplementary light duration on the activities of four antioxidant enzymes (superoxide dismutase, SOD; catalase, CAT; ascorbate peroxidase, APX and guaiacol peroxidase, GPX) in seedlings of two cultivars of watermelon, 'Speed' and 'Sambok Honey', grafted onto 'RS Dongjanggun' bottle gourd rootstocks, after 10 days of supplementary light treatments. The error bars represent the SEs of the biological replicates ($n = 3$). The mean separation within columns are significantly different according to Tukey's multiple range test at $p = 0.05$.

## 4. Discussion

Through analyzing the results, it was found that most supplementary light treatments yielded seedlings with much better growth and development than the control (CK). With an increased duration of supplementary light, the length of scions tended to increase. However, it is also a bad phenomenon if the scion is only greater in length with slow accumulation of biomass. During the seedling cultivation period, stretchy growth of seedlings caused by insufficient light is very common [46,47]. Many seedling growers use the plant growth regulators (PGRs) such as chlormequat chloride, paclobutrazol, uniconazole, etc. which are banned by the Korean government, to inhibit the stretchiness of seedlings. Though effective, it is not the fundamental way to solve the problem.

The fresh and dry weights of the scions, rootstocks, and roots of seedlings grown with 12 and 16 h of supplementary lighting had no significant difference between them, but were significantly higher than those of seedlings grown with 8 h of or no supplementary lighting. Jiang et al. [13], Li and Kubota [48] etc. also confirmed that proper supplementary light can increase the biomass of crops.

In this study, supplementary light also increased the diameter of scions and rootstocks. Klamkowski, Ochieng, and Piszczek [11,49,50] also described that a lack of light caused plants to slender and stretchy, become etiolated and develop thin stems, and were more susceptible to damage and diseases.

Compared with those in CK, seedlings grown with supplementary light had a higher number and greater area of leaves. Seedlings grown with 12 and 16 h of supplementary light had no significant differences in the number and area of leaves. The increase in the number and area of leaves demonstrates that supplementary lighting accelerated the growth and development of seedlings. Leaves of high-quality seedlings should be thick and robust. Hence, the specific leaf weight was calculated to measure the quality of the leaves. It was found that seedlings in CK had the lowest specific leaf weight. 'Sambok Honey' seedlings grown with 12 h of supplementary light had the greatest specific leaf weight. Seedlings of both cultivars had the highest chlorophyll content when grown with 12 and 16 h of supplementary light. This indicates that supplementary lighting not only promoted the biomass accumulation of watermelon seedlings, but also improved the quality of leaves. Hao and Papadopoulos [51] also found that supplementary lighting promoted the biomass accumulation and leaf quality in cucumber seedlings.

Growers often say that the quality of the roots is very important in the seedling industry [52–54]. The quality of the roots directly affects the growth and development of seedlings after transplanting, thus affecting the quality and yield of crops. After being treated with supplementary light, it was observed that both the biomass and the length of roots increased. Figure 4 shows that both cultivars of watermelon seedlings grown with 12 and 16 h of supplementary light had more lateral roots than those with 8 h of supplementary light and in CK. Seedlings grown in CK had very weak roots, which made it difficult to pull them out of the plug tray together with the medium. It could also be seen from Figure 2 that the dry weight ratio of the shoot to root of seedlings grown with 12 and 16 h of supplementary light was the lowest, indicating that supplementary light promoted the growth of underground plant parts more than it did the above-ground plant parts.

The total biomass was calculated by adding the dry weights of the scion, cotyledon and stem of the rootstock and roots. It was found that the seedlings grown with 12 or 16 h of supplementary light had the greatest biomass without much difference between them, while seedlings in the CK had the lowest biomass. The average seedling biomass of the two watermelon cultivars supplemented with 12 or 16 h of light had no significant difference. Through the common light response curves for photosynthesis in plant, we know that the light compensation point is the minimum light intensity at which the organism shows a gain of carbon fixation. The net photosynthetic rate shows a linear rise in response to increased light, in the range of light limitation. At higher light levels, saturation occurs as the efficiency of the photosynthetic mechanism is reduced due to the activation of energy quenching processes. Under excess light conditions, net photosynthesis can decline as a result of photoxidative stress [55]. Velez-Ramirez et. al [56] reviewed the response of plants under continuous light and mentioned that continuous light induced photo-oxidative damage in some species. It could induce carbon unbalance which led to down-regulated photosynthesis. Long photoperiod had negative effects on growth in tomato plants [57]. These are inferred as the reasons that the biomass of 16 h of supplementary light did not increase significantly as compared to that in the 12 h in this study.

Based on this inference, we performed abiotic stress analysis. Compared to the seedlings grown with 8 h of or no supplementary light, the average biomass of seedlings grown with 16 h of supplementary light was 34.0% and 66.0% higher, respectively. This shows that the duration of the supplementary light was not directly proportional to the biomass accumulated through photosynthesis. When the quantity of light reached a certain limit, the photosynthetic rate no longer increased or even was suppressed. Garland et. al [58] also found that as the DLI increased, the number of leaves, leaf area, and shoot dry weight of *Heuchera americana* all had the tendency of increasing to peak and then decreasing. Earlier studies have shown that the best growth and yield of tomatoes were obtained under a photoperiod of 14 h. Longer photoperiods did not increase the yield. 20 or 24 h photoperiods could even decrease the yield and caused leaf chlorosis after 6 to 8 weeks [59,60].

The scion dry weight to height ratio is another indicator of the seedling quality. The higher the ratio, the shorter, stronger, and more compact the seedlings are. A high scion dry weight to height ratio also shows that there was no overgrowth. Figure 2 shows that the ratio of the scion dry weight to height of 'Sambok Honey' seedlings grown with 8 h of supplementary light, as well as 'Sambok Honey' and 'Speed' seedlings grown with 12 and 16 h of supplementary light were the greatest. These results show that while supplementary light improved the quality of seedlings, 8 h of supplementary light was not enough to most effectively improve the seedling quality.

Dickson's quality index and the use efficiency of supplementary light were calculated in order to better compare the differences between the treatments (Figure 3). The Dickson's quality index was originally used to measure the quality of woody plants and has since been widely used in herbaceous crops. In Bantis et. al's study, the Dickson's quality index was used as an important indicator for grafted watermelon and squash seedlings [43,61]. Neto et. al also mentioned that Dickson's quality index is a good indicator for characterization of quality of eggplant seedlings [62]. Two cultivars of watermelon seedlings grown with 12 and 16 h displayed the greatest Dickson's quality index, without a significant difference between them. The average supplementary light use efficiency of the two cultivars of watermelon seedlings was 23% lower at 16 h than at 12 h. From an energy savings perspective, 12 h of supplementary lighting was the most effective in improving the growth and quality of the two cultivars of grafted watermelon seedlings.

The contents of carbohydrates and soluble proteins reflect the biomass accumulation of seedlings from another perspective. Sugars produced by photosynthesis of most plants are sucrose firstly, which is quickly transformed into starch. They are temporarily stored in the chloroplast, and then transported to various parts of the plant. The plant can also synthesize proteins, lipids, and other organic matters. These carbohydrates reflect the efficiency of photosynthesis. In Figure 5, it can be seen that contents of soluble sugar, starch and protein of watermelon seedling treated with supplementary light were significantly higher than those of the CK, and the accumulation tends to increase with the increasing duration of supplementary lighting. No significant difference between 12 and 16 h suggests that assimilation peaked after 12 h of supplementary lighting. Erhioui et. al [63] found that by supplementing light of tomato plants grown in the greenhouse, the accumulation of carbohydrate, especially starch, was significantly increased. Hidaka et. al [64] treated the strawberries in the greenhouse with LED supplementary light and found that 12 h of supplementary light treatment showed a significant increase in leaf photosynthesis, earlier differentiation of flower buds on the second inflorescence, and the accumulation of carbohydrates in the fruit.

Based on the above research results, we can almost suggest that 12 h is the optimal supplementary light duration for grafted watermelon seedlings before the transplanting in winter in South Korea. To further support this conclusion, we analyzed the activities of several related enzymes of these two cultivars of watermelon seedlings treated with different supplementary light durations from the perspective of plant abiotic stress. Both insufficient and excessive light can cause abiotic stresses to plants [65–68]. Hydrogen peroxide ($H_2O_2$) is a reactive oxygen species in the metabolism of plant cells. It is also an important indicator that signals that a plant cell is under stress. Measurements of the $H_2O_2$ content in seedlings revealed that 16 h of supplementary lighting resulted in the highest $H_2O_2$ content, indicating that it was excessive to the two cultivars of watermelon at the seedling stage. Müller's [69] research showed that a high level of light, especially ultraviolet (UV) light, caused stresses that could potentially lead to serious damages to the DNA, proteins, and other cellular components, and even suggested that plants might develop an epigenetic memory of UV and light stresses. Suzuki et. al [70] uncovered alterations in the mRNA levels in plants acclimating to light stresses. They showed that more than 20% of transcripts accumulating in plants within 20–60 s of initiation of light stresses were $H_2O_2$ and ABA-response transcripts. This indicates that plants very quickly respond to stresses, producing $H_2O_2$. Neill and Desikan [71] summarized the regulation mechanism of $H_2O_2$ in plants. They mentioned that antioxidant systems could normally balance the $H_2O_2$ production in plant cells [72,73], but abiotic stresses such as dehydration, low and high temperatures, and excessive

irradiation could disturb this balance, such that increased $H_2O_2$ levels initiated signaling responses that included enzyme activation, gene expression, programmed cell death (PCD) and cellular damage. The activity of the antioxidant enzymes were measured and it was found that the activities of the four enzymes (SOD, CAT, APX, and GPX) were indeed the highest when seedlings were grown with 16 h of supplementary lighting. Generally, the antioxidant enzymes prevented the cell damages caused by the reactive oxygen species (ROS) by catalyzing a cascade of reactions. As the first line of defense, SOD dismutased the $O_2^{1-}$ radical into $H_2O_2$ and molecular oxygen. Subsequently, the generated $H_2O_2$ was eliminated by the action of CAT, APX and GPX ($H_2O_2$ scavengers) [74]. The increased activities of these enzymes also further suggested that the seedlings grown with 16 h supplementary light were growing under certain stresses. However, whether the enhanced antioxidant mechanism at 16 h supplementary lighting might benefit the seedlings after transplanting in the field remains to be studied.

## 5. Conclusions

In general, supplementary light is very effective in improving the quality of watermelon seedlings. Studying the effects of different durations of supplementary lighting, it was observed that 12 h of supplementary light (Supplementary DLI = 4.32 $mol \cdot m^{-2} \cdot d^{-1}$) not only improved the growth than CK and 8 h treatments, but also had lower activities of antioxidant enzymes than 16 h of treatment for the grafted watermelon 'Speed' and 'Sambok Honey' seedlings. Overall, 12 h was suggested as the most effective supplementary lighting duration in improving the quality of the two cultivars of grafted watermelon seedlings. In practice, even if natural light varies due to regional or environmental differences, a DLI of about 14 $mol \cdot m^{-2} \cdot d^{-1}$ (natural + supplemental light) is recommended for improving the quality of watermelon seedlings. It is the authors' wishes that these findings provide some reference value for plug seedling growers in improving the product quality and production efficiency.

**Author Contributions:** Conceptualization, B.R.J. and H.W.; methodology, B.R.J. and H.W.; software, H.W. and M.W.; validation, B.R.J.; formal analysis, B.R.J. and H.W.; investigation, H.W. and M.W.; resources, B.R.J.; data curation, H.W. and M.W.; writing—original draft preparation, H.W.; writing—review and editing, B.R.J. and H.W.; visualization, H.W. and M.W.; supervision, B.R.J.; project administration, B.R.J. and H.W.; funding acquisition, B.R.J. and H.W. All authors have read and agreed to the published version of the manuscript.

**Funding:** This research was funded by Korea Institute of Planning and Evaluation for Technology in Food, Agriculture, Forestry and Fisheries, Rural Development Administration, Korea (Project No.319008-01), and Cooperative Research Program for Agriculture Science and Technology Development (Project No. PJ012773022018). H.W. and M.W. were supported by a scholarship from the BK21 Plus Program, Ministry of Education, Korea.

**Acknowledgments:** This study was carried out with support from the Institute of Planning and Evaluation for Technology in Food, Agriculture, Forestry and Fisheries, Rural Development Administration, Korea (Project No.319008-01), and Cooperative Research Program for Agriculture Science and Technology Development (Project No. PJ012773022018). H.W. and M.W. were supported by a scholarship from the BK21 Plus Program, Ministry of Education, Korea.

**Conflicts of Interest:** The authors declare no conflict of interest.

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
