# Peer review of "Effect of Supplementary Lighting Duration on Growth and Activity of Antioxidant Enzymes in Grafted Watermelon Seedlings"

_agronomy, doi:10.3390/agronomy10030337_

Round 1

Reviewer 1 Report

Comments for Agronomy

Manuscript Number: agronomy-714105

Title: Effect of Supplementary Light Duration on Growth 2 and Development of and Activity of Antioxidant 3 Enzymes in Grafted Watermelon Seedlings

The authors tested the effect of three supplemental LED light durations (photoperiods) or no supplemental light on the quality of two varieties of grafted watermelon seedlings during greenhouse cultivation. They conducted morphological as well as phytochemical determinations which sufficiently depict watermelon seedling quality.

The Introduction section is very informative and sufficiently describes the scope of the research. In the Materials and methods section, the setup is properly described but some more details should be included. The Discussion section is clear and gives satisfying explanations regarding the light duration effects. However, the Conclusion should be improved.

In general, the manuscript is well written and the English are good. Specific comments are the following.

Title: The title should be amended since there are syntax issues. E.g. Effect of Supplementary Light Duration on Growth, Development, and Activity of Antioxidant Enzymes in Grafted Watermelon Seedlings.

Keywords: antioxidant enzyme should be omitted since it is referred to in the title.

Introduction

Lines 55-64. Please move this paragraph after the next one (after line 75), because the “light” paragraphs are interrupted by the “LED” paragraph.

Line 65. This is a repetition of lines 46-47. Please amend or omit the sentence.

Lines 74-75. Please rephrase this sentence.

Lines 78-81. Are these data published? If not, please add “unpublished data”.

Lines 87-88. These two articles (19,20] deal with grafted tomato seedlings and optimal light quality and intensity might differ from watermelon. You should also refer to watermelon seedlings.

Materials and methods

Lines 93-102. Please provide some information about the scion and rootstock production (e.g. type of greenhouse, environmental conditions, substrate, etc) and the healing stage of grafted seedlings.

Line 119. How did you ensure that the treatments (CK and LED durations) did not interfere with each other?

Line 175. There is some kind of error in this line. Probably a misplaced dot. Please check and correct it.

Line 185. Did you have any problems when cleaning the substrate from the roots? Depending on the substrate, it might difficult to clear them properly, and some roots may be accidentally cut off.

Results

Figure 4. This picture shows the differences between treatments, but it would be positive if you have another one where seedlings are more widely arranged.

Discussion

Line 340. Please use the whole term for plant growth regulators (PGR) and other abbreviations the first time you refer to it.

Line 405-406. This part is only a repetition of lines 269-270. Could you please elaborate on these results because DQI is a quality index for grafted watermelon seedlings?

Lines 425-426. From the abovementioned results (growth and development) it is clear that 12 h are sufficient for the production of high-quality grafted watermelon seedlings. However, the enhanced antioxidant mechanism at 16 h lighting might benefit the seedlings after transplanting in the field. I think you should not draw a definite conclusion unless you can support it with references.

Conclusions

This section can be improved. Apart from your main finding, you should briefly describe why you think 12 h of supplementary lighting is better than 16 h. Moreover, you should describe why 0 h and 8 h are inadequate for grafted watermelon seedling production.

In-text references should be correctly numbered in the whole manuscript. For example, in line 65, you mention reference [29], but it should be [19].

Author Response

Title: The title should be amended since there are syntax issues. E.g. Effect of Supplementary Light Duration on Growth, Development and Activity of Antioxidant Enzymes in Grafted Watermelon Seedlings.

Another reviewer's recommendation:

I suggest shortening the title: Effect of supplementary lighting duration on growth and activity of antioxidant enzymes in grafted watermelon seedlings. It is no need to use: “development of”  (the experiment concerns vegetative growth of seedlings).

After comprehensive consideration, the results of development in the manuscript are very few, So, we removed the ‘development’.

Keywords: antioxidant enzyme should be omitted since it is referred in the title.

It was removed.

Introduction

Lines 55-64. Please move this paragraph after the next one (after line 75), because the “light” paragraphs are interrupted by the “LED” paragraph.

It was moved.

Line 65. This is a repetition of lines 46-47. Please amend or omit the sentence.

This sentence was removed.

Lines 74-75. Please rephrase this sentence.

The sentence was revised.

Lines 78-81. Are these data published? If not, please add “unpublished data”.

The (unpublished data) was added after this sentence

Lines 87-88. These two articles (19,20] deal with grafted tomato seedlings and optimal light quality and intensity might differ from watermelon. You should also refer to watermelon seedlings.

References [69-71] dealing watermelon seedlings were added.

Materials and methods

Lines 93-102. Please provide some information about the scion and rootstock production (e.g. type of greenhouse, environmental conditions, substrate etc) and the healing stage of grafted seedlings.

The type of greenhouse, environmental conditions, substrate used were added in materials and methods.

Line 119. How did you ensure that the treatments (CK and LED durations) did not interfere with each other?

Enough space was left to ensure that the treatments (CK and LED durations) did not interfere with each other. It was described in materials and methods now.

Line 175. There is some kind of error in this line. Probably a misplaced dot. Please check and correct it.

It was corrected, and ‘the seedlings grow’ was removed.

Line 185. Did you have any problems when cleaning the substrate from the roots? Depending on the substrate it might difficult to properly clear them and some roots may be accidentally cut off.

This mixture we used was a mixture of peat moss and perlite, and it was easy to wash off by using a shower head with a proper water pressure not to damage the root.

Results

Figure 4. This picture shows the differences between treatments but it would be positive if you have another one where seedlings are more widely arranged.

Unfortunately, in order not to make the photo look too wide with not much more information, the plants were arranged a little closely each other. We do not have photos with more widely arranged plants. This is a valuable suggestion. We will pay attention to this point in our follow-up studies.

Discussion

Line 340. Please use the whole term for plant growth regulators (PGR) and other abbreviations the first time you refer to it.

The full name has been added.

Line 405-406. This part is only a repetition of lines 269-270. Could you please elaborate on these results because DQI is a quality index for grafted watermelon seedlings?

References indicating that DQI is a quality index for grafted watermelon seedlings were added.

Lines 425-426. From the abovementioned results (growth and development) it is clear that 12 h are sufficient for the production of high-quality grafted watermelon seedlings. However, the enhanced antioxidant mechanism at 16 h lighting might benefit the seedlings after transplanting in the field. I think you should not draw a definite conclusion unless you can support it with references.

It was highlighted by adding ‘before the transplanting’ in this sentence. And we added a phrase ‘whether the enhanced antioxidant mechanism at 16 hours supplementary lighting could benefit the seedlings after transplanting in the field remains to be studied’ at the end of the discussion.

Conclusions

This section can be improved. Apart from your main finding, you should briefly describe why you think 12 h of supplementary lighting is better than 16 h. Moreover, you should describe why 0 h and 8 h are inadequate for grafted watermelon seedling production.

The description has been added:

12 hours of supplementary light (Supplementary DLI = 4.32 mol·m-2·d-1) not only improved the growth than 0 and 8 hours treatments, but also had lower activities of antioxidant enzymes than 16 hours of treatment for the grafted watermelon ‘Speed’ and ‘Sambok Honey’ seedlings.

In text references should be correctly numbered in the whole manuscript. For example, in line 65 you mention reference [29] but it should be [19].

The sentence of line 65 was removed.

Reviewer 2 Report

The work is focused on an interesting issue which is currently new in literature:  using supplementary LED light for improving the quality of grafted seedlings of two watermelon cultivars growing in greenhouse. The authors proposed for plug seedling growers an original reference DLI value recommended for watermelon seedling production with consideration of the energy consumption. The work is valuable from the scientific point of view and horticulture practice.

In my opinion the manuscript requires several significant changes and additions; after the revision will be suitable for publication in the Agronomy.  

Detailed comments:

  • I suggest shortening the title: Effect of supplementary light duration on growth and activity of antioxidant enzymes in grafted watermelon seedlings. It is no need to use: “development of” (the experiment concerns vegetative growth of seedlings).   
  • Lines 38, 81, 456-458: The Authors should present DLI for each experimental light treatment, preferably in Material and Methods p. 2.2. Providing DLI value in Conclusions (and only for one light treatment) is insufficient. The discussion would be substantially better with reference to DLI values used in experiment.
  • Lines 55-57: It would be better to cite more recently works available in actually literature on LED light in horticulture.
  • Lines 50 and 65: Please arrange the numerical order of citations at the work.
  • MM: when the experiment was conducted? The term ‘winter’ is too general.
  • Line 104: The light intensity is depended on distance between light source and a place of light measurement. It would be good give some another information here: dimensions of the LED lamps, energy consumption and distance from the lamps to obtain PPFD 100 µmol m-2 s-1 on the plants level.
  • Figure 1.: The vertical axis is incorrectly described. It should be (without units): Relative intensity.
  • Line 138, 142: Instead the term ‘soluble sugars’ or ‘starch’ it should be ‘extract’ or ‘plant extract’ (in which these compounds will be detected).
  • Lines 175-177: Please correct this sentence.
  • Lines 319-320: This relationship is only true for the Speed variety.
  • Lines 321-322 – but I see significant differences...

Lines 345-349: Please correct the citations in these sentences: Jiang et al. [32], Li and Kubota [40] etc.

Author Response

  • I suggest shortening the title: Effect of supplementary light duration on growth and activity of antioxidant enzymes in grafted watermelon seedlings. It is no need to use: “development of” (the experiment concerns vegetative growth of seedlings).
  • It waschanged as recommended.
  • Lines 38, 81, 456-458: The Authors should present DLI for each experimental light treatment, preferably in Material and Methods 2.2. Providing DLI value in Conclusions (and only for one light treatment) is insufficient. The discussion would be substantially better with reference to DLI values used in experiment.
  • The supplementary DLI of all treatment was added in Material and Methods.
  • Lines 55-57: It would be better to cite more recently works available in actually literature on LED light in horticulture.
  • Several recent articles was added.
  • Lines 50 and 65: Please arrange the numerical order of citations at the work.
  • Because references were added and deleted during the manuscript review period, we will rearrange them after the final modification.
  • MM: when the experiment was conducted? The term ‘winter’ is too general.
  • The entire experimental period ‘December 15th- 25th of 2018’ was added in the MM.
  • Line 104: The light intensity is depended on distance between light source and a place of light measurement. It would be good give some another information here: dimensions of the LED lamps, energy consumption and distance from the lamps to obtain PPFD 100 µmol m-2 s-1 on the plants level.
  • ‘One LEDs bar (150x3 cm) was series connected by 200 LED chips and the power of each LED chip was 2 watts’ and ‘The distance from the top leaf of the seedlings was about 85 cm’ was added in the MM.
  • Figure 1.: The vertical axis is incorrectly described. It should be (without units): Relative intensity.
  • Light intensity with current unit is the right way to described since it is not a relative intensity, but absolute intensity.
  • Line 138, 142: Instead the term ‘soluble sugars’ or ‘starch’ it should be ‘extract’ or ‘plant extract’ (in which these compounds will be detected).
  • The extract was added.
  • Lines 175-177: Please correct this sentence.
  • This sentence was corrected.
  • Lines 319-320: This relationship is only true for the Speed variety.
  • The description for ‘Sambok Honey’ was added.
  • Lines 321-322 – but I see significant differences.
  • Itwas
  • Lines 345-349: Please correct the citations in these sentences: Jiang et al. [32], Li and Kubota [40] etc.
  • This sentence was corrected.